# Algorithmic fairness with monotone likelihood ratios

**Wes Camp**                                                                    *wesacamp@gmail.com*
*Optum*

**Reviewed on OpenReview:** *https://openreview.net/forum?id=mtoWaOgIKy*

## Abstract

We show that inequalities of many commonly used fairness metrics (true/false positive/negative rates, predicted positive/negative rates, and positive/negative predictive values) are guaranteed for groups with different outcome rates under a monotonically calibrated model whose risk distributions have a monotone likelihood ratio, extending existing impossibility results. We further provide lower bounds on the FNR/FPR disparities and PPR/PNR disparities in the same setting, showing that either the FNR disparity or FPR disparity is at least as large as the positive outcome rate disparity (for FNR disparity) or negative outcome rate disparity (for FPR disparity), and either the PPR disparity or PNR disparity is at least as large as the positive outcome rate disparity (for PPR disparity) or negative outcome rate disparity (for PNR disparity). While incompatibilities of some combinations of these metrics have been demonstrated previously, we are unaware of any work that has demonstrated direct incompatibility of calibration with these individual equalities, equivalence of these inequalities, or lower bounds for the disparity in these values under distributional assumptions about a model's predictions.

## 1  Introduction

In 2016, ProPublica published an analysis (Angwin et al., 2016) of the COMPAS recidivism prediction algorithm, which assesses a defendant's risk of committing another crime and is used to assist judges in making pre-trial release and sentencing decisions. Using publicly available data on the algorithm's predictions and outcomes in Broward County, Florida, they determined that the false positive rate for Black defendants was significantly higher than for White defendants, i.e. a non-recidivating Black defendant was almost twice as likely to be asssessed as high-risk than a non-recidivating White defendant. The false negative rate for White defendants was also significantly higher than for Black defendants, meaning a recidivating White defendant was much more likely to be incorrectly labeled as low risk than a recidivating Black defendant. Based on these results, they concluded that the algorithm was biased against Black defendants. Northpointe, the developer of the COMPAS software, rebutted that the algorithm is fair because it satisfies predictive parity (equal positive predictive values) across the two groups (Dieterich et al., 2016). Shortly after, both Chouldechova (2017) and Kleinberg et al. (2017) established that these notions of fairness (equal false positive/negative rates and predictive parity) are impossible to satisfy simultaneously, except in trivial cases. Kleinberg et al. (2017) further prove the impossibility of well-calibration (scores equaling the outcome probabilities across groups) and balance for the positive/negative class (equality of average scores for members in the positive/negative class). Pleiss et al. (2017) demonstrated additional conflict between error rate equality and well-calibration by showing that well-calibration is compatible with only a single error rate constraint, but achieving this compatibility essentially requires randomizing predictions from an existing classifier, an intuitively disturbing procedure.

While well-calibration is clearly a desirable condition for both the fairness and effectiveness of such an algorithm (Pleiss et al., 2017; Corbett-Davies et al., 2023), the same is not necessarily true of false positive/negative rate equality. Many authors have identified both technical and social issues resulting from reliance on these metrics (Corbett-Davies et al., 2023; Chohlas-Wood et al., 2023; Pierson, 2024), including "the problem of infra-marginality" (Ayres, 2002; Simoiu et al., 2017), the sensitivity of these metrics to

model behavior and population distributions far away from the decision margin. Nonetheless, they are still commonly relied upon by machine learning practitioners for detecting potential bias in models. For example, many popular algorithmic fairness toolkits, including Microsoft's Fairlearn (Bird et al., 2020), IBM's AI Fairness 360 (Bellamy et al., 2019), and Aequitas (Saleiro et al., 2019), calculate ratios of these metrics across groups as part of their fairness reports, with some identifying the ratios as the *disparity* in the corresponding metric. Some of these further instruct users that these disparities and/or the disparities in predicted positive/negative rates must be between 80% and 125% in order for the model to be fair, a range derived from disparate impact law in a process that has been deemed "epistemic trespassing" (Watkins & Chen, 2024). If the toolkit or end user determines that a given disparity between two groups is sufficiently large, mitigation algorithms (such as those described by Agarwal et al. (2018) and Hardt et al. (2016)) are provided that can adjust the model's predictions or choose group-specific thresholds to achieve a specified level of parity, at the cost of (potentially substantial) performance degradation at both the population and group level (Pfohl et al., 2021).

When an algorithm satisfies calibration across two groups (where predictions correspond to equal outcome rates across groups, but not necessarily the exact outcome probabilities), differences in these metrics are direct consequences of differences in group risk distributions under the algorithm. In what follows, we examine scenarios where these two risk distributions satisfy the *monotone likelihood ratio property* (MLRP), a frequently considered property in statistical problems that is satisfied by many commonly encountered families of distributions, including the exponential, binomial, Poisson, or normal with a fixed $\sigma$ (Karlin & Rubin, 1956). Any two beta or gamma distributions that cross exactly once also satisfy the MLRP (Gaebler & Goel, 2025), and the property is preserved under monotone transformations (e.g. the above statements also hold for the corresponding log-families) and sums, if the distributions are log-concave (see Shaked & Shanthikumar (2007) for details of these and other properties). Gaebler & Goel (2025) considered distributions satisfying the MLRP in the context of testing outcomes for fairness, and they provide empirical evidence that the property holds in domains including recidivism, loan approvals, police searches, and law school admissions, by demonstrating that predictive models developed on data from those domains have group risk distributions satisfying the property. Chiang et al. (2025) proposed a method for developing disease progression models that account for systemic disparities, using the MLRP to reason about the effects of differences in initial severity, rates of progression, or visit frequency across groups on model outcomes. The property has also been studied in contexts such as measuring treatment effects in randomized control trials (Chemla & Hennessy, 2019) and identifying racial disparities in police searches (Anwar & Fang, 2006; Feigenberg & Miller, 2022). While the MLRP is often assumed as a property of the underlying risk distributions, an assumption that is difficult or impossible to verify in many situations, our results only require that the risk distributions under the examined model satisfy the property. This has the advantages of being both easily verifiable and at least equally reasonable, as a sufficiently appropriate model will approximate the true risk distributions.

The contributions of this paper are as follows: we introduce a slightly stronger version of calibration (but still significantly weaker than well-calibration) called monotonic calibration, which requires increasing model predictions to correspond to increasing outcome probabilities, but not the exact outcome probabilities as required by well-calibration. We then show that if a model is monotonically calibrated for two groups and the risk distributions of the groups under the model satisfy the MLRP, then for any non-trivial decision threshold, the group with the higher outcome rate will have:

- a larger predicted positive rate (equivalently, a smaller predicted negative rate),

- a larger false positive rate and smaller false negative rate,

- a larger positive predictive value (unless the likelihood ratio or outcome probabilities are constant above the decision threshold, in which case the PPVs are equal), and

- a smaller negative predictive value (unless the likelihood ratio or outcome probabilities are constant below the decision threshold, in which case the NPVs are equal).

We further show that either:

- the disparity in predicted positive rates is greater than the disparity in positive outcome rates, or

- the disparity in predicted negative rates is greater than the disparity in negative outcome rates

and also that either

- the disparity in false negative rates is greater than the disparity in positive outcome rates, or

- the disparity in false positive rates is greater than the disparity in negative outcome rates.

In particular, for many commonly encountered distributions of model predictions, calibration is incompatible with equality of any of these metrics when outcome rates are unequal, and even approximate equalized odds (equal false negative and false positive rates) and approximate demographic parity cannot be satisfied between groups with large differences in outcome rates. We conclude with an example of how these results guarantee large disparities in the error rates studied in the COMPAS dataset, and discuss the dangerous consequences of reliance on these metrics for detecting bias.

## 2 Main Results

In what follows we discuss a model $M$ intended to predict the probability of a binary outcome, but we do not require that $M$ take values only in $[0, 1]$, so $M$ may be a more general risk score (as in the COMPAS example).

**Definition 1.** Let $M$ be a model defined on groups $P$ and $Q$ and used to predict the probability of a binary outcome $O(r) : P \cup Q \rightarrow \{0, 1\}$ for $r \in P \cup Q$. $M$ is *calibrated* for groups $P$ and $Q$ if for each predicted value $x$

$$\Pr(O(p_i) = 1 \mid p_i \in P, M(p_i) = x) = \Pr(O(q_i) = 1 \mid q_i \in Q, M(q_i) = x)$$

and $M$ is *well-calibrated* for $P$ and $Q$ if for each predicted value $x$

$$\Pr(O(p_i) = 1 \mid p_i \in P, M(p_i) = x) = \Pr(O(q_i) = 1 \mid q_i \in Q, M(q_i) = x) = x.$$

The term *calibration* is often used to mean *well-calibration* as given above. We distinguish between the two properties here because well-calibration is not necessary for the results that follow; however, we do require a slightly stronger version of calibration, that ensures that increasing predictions correspond to increasing outcome rates. This avoids degenerate cases where predictions do not correspond to outcomes in any natural way. Such cases rarely occur in practice, and most models satisfying (not necessarily well-)calibration for groups will also satisfy this stronger property.

**Definition 2.** Let $M$ be a model defined on groups $P$ and $Q$ and used to predict the probability of a binary outcome $O(r) : P \cup Q \rightarrow \{0, 1\}$ for $r \in P \cup Q$. $M$ is *monotonically calibrated* for groups $P$ and $Q$ if it is calibrated for those groups, and the function $c : \mathrm{Range}(M) \rightarrow [0, 1]$ given by

$$c(x) = \Pr(O(p_i) = 1 \mid p_i \in P, M(p_i) = x) = \Pr(O(q_i) = 1 \mid q_i \in Q, M(q_i) = x)$$

is increasing and not almost everywhere constant.

For what follows we treat a model $M$ as a real-valued random variable on a probability space $P \cup Q$, and assume the cumulative distribution functions $F_P(x) = \Pr(M(p_i) \leq x \mid p_i \in P)$ and $F_Q(x) = \Pr(M(q_i) \leq x \mid q_i \in Q)$ are absolutely continuous with associated density functions $p(x)$ and $q(x)$.

**Definition 3.** Two probability density functions $p(x)$ and $q(x)$ satisfy the *monotone likelihood ratio property (MLRP)* with $p$ dominating, denoted $p \succeq_{lr} q$, if for almost every pair $x_0 < x_1$, $\dfrac{p(x_0)}{q(x_0)} \leq \dfrac{p(x_1)}{q(x_1)}$. We denote $p \succ_{lr} q$ if $p \succeq_{lr} q$ but $q \not\succeq_{lr} p$, or equivalently if $p \succeq_{lr} q$ and $\dfrac{p(x)}{q(x)}$ is not constant almost everywhere.

Note that if $p \succeq_{lr} q$, then $p$ first-order stochastically dominates $q$. See Shaked & Shanthikumar (2007) for more general properties of distributions satisfying the MLRP.

**Lemma 1.** *Let $p(x)$ and $q(x)$ be the probabilty density functions of the predictions of a model $M$ on groups $P$ and $Q$, and let $O(P) = Pr(O(p_i) = 1 \mid p_i \in P)$ and $O(Q) = Pr(O(q_i) = 1 \mid q_i \in Q)$ denote the binary outcome rates for groups $P$ and $Q$. If $M$ is monotonically calibrated for groups $P$ and $Q$ (via function $c(x)$) and $p$ and $q$ satisfy the MLRP (with either $p \succeq_{lr} q$ or $q \succeq_{lr} p$) then $O(P) > O(Q)$ iff $p \succ_{lr} q$.*

*Proof.* ($\Leftarrow$) Letting $c_0 = \inf_x c(x) \geq 0$ and using that $c(x)$ and $p(x)/q(x)$ are increasing and not almost everywhere constant, we must have both $c(x) > c_0$ and $p(x) > q(x)$ on some set of non-zero measure, so we have $O(P) - O(Q) = \int c(x)(p(x) - q(x))dx > \int c_0(p(x) - q(x))dx = 0$.

($\Rightarrow$) From the proof of the direction above, if $q \succ_{lr} p$ then $O(Q) > O(P)$, so we must have that $p \succeq_{lr} q$. If $p(x)/q(x)$ is constant almost everywhere then using that $\int p(x) = \int q(x) = 1$ we obtain $O(P) = \int c(x)p(x) = \int c(x)q(x) = O(Q)$, thus $p \succ_{lr} q$. $\qquad\square$

**Definition 4.** The *predicted positive rate* $PPR_{M,t,P}$ of a model $M$ with decision threshold $t$ (so that a prediction less than $t$ is considered a predicted negative, and a prediction at least $t$ is considered a predicted positive) on a group $P$ is the proportion of $P$ with predicted value at least $t$. The *predicted negative rate* $PNR_{M,t,P} = 1 - PPR_{M,t,P}$ is the proportion of $P$ with predicted value less than $t$.

The predicted positive rate defined above is sometimes called *selection rate*, *admission rate*, or various other terms, depending on the application of the model under discussion. Equality of these rates across groups is also know by various terms including *demographic parity*, *group fairness*, and *statistical parity*. The following lemma demonstrates that inequality of these rates is an immediate consequence of (strict dominance under) the MLRP, without any assumptions about the calibration of the model.

**Lemma 2.** *Let $p(x)$ and $q(x)$ be the probabilty density functions of the predictions of a model $M$ on two groups $P$ and $Q$. Let $t$ be a decision threshold for the predictions of $M$. If $p \succ_{lr} q$ then $PPR_{M,t,Q} \leq PPR_{M,t,P}$ (with equality only if both are 0 or 1) and $PNR_{M,t,P} \leq PNR_{M,t,Q}$ (with equality only if both are 0 or 1).*

*Proof.* The desired inequality is:

$$PPR_{M,t,Q} = \int_{x \geq t} q(x)dx \leq \int_{x \geq t} p(x)dx = PPR_{M,t,P}$$

which follows directly from the first-order stochastic dominance of $p$ over $q$. To obtain strictness of the inequality when both are not 0 or 1, note that if $\int_{x \geq t} q(x)dx = \int_{x \geq t} p(x)dx$, integrating both sides of the MLRP inequality $p(x_0)q(x_1) \leq p(x_1)q(x_0)$ for $x_0 \leq t \leq x_1$ over the region $x_1 \geq t$ gives $q(x) \geq p(x)$ on $x < t$. But since $\int q(x)dx = \int p(x)dx = 1$, we have $\int_{x < t} q(x)dx = \int_{x < t} p(x)$, implying $q(x) = p(x)$ almost everywhere on $x < t$. Integrating both sides of $p(x_0)q(x_1) \leq p(x_1)q(x_0)$ over the region $x_0 < t$ gives $p(x) \geq q(x)$ on $x \geq t$, which then implies $q(x) = p(x)$ almost everywhere on $x \geq t$, contradicting that $p \succ_{lr} q$. $\qquad\square$

As the $PPR$ for any group must approach 1 as the decision threshold $t$ decreases, the $PPR$ disparity across groups is necessarily small for sufficiently small $t$. Similarly, the $PNR$ disparity is necessarily small for sufficiently large $t$. However, the next lemma shows that the MLRP and monotonic calibration guarantee that for a non-trivial decision threshold, one of these disparities is at least as large as the positive outcome rate disparity (for $PPR$) or negative outcome rate disparity (for $PNR$).

**Lemma 3.** *Let $p(x)$ and $q(x)$ be the probabilty density functions of the predictions of a model $M$ on two groups $P$ and $Q$. If $M$ is monotonically calibrated for groups $P$ and $Q$ (by a function $c(x)$) and $p \succ_{lr} q$, then for any decision threshold $t$ where the ratios below are defined, we have either*

$$\frac{PPR_{M,t,P}}{PPR_{M,t,Q}} \geq \frac{O(P)}{O(Q)}$$

*or*

$$\frac{PNR_{M,t,Q}}{PNR_{M,t,P}} \geq \frac{1 - O(Q)}{1 - O(P)}.$$

*Proof.* Using that $M$ is monotonically calibrated the first inequality is

$$\int_{x \geq t} p(x)dx \int c(x)q(x)dx \geq \int_{x \geq t} q(x)dx \int c(x)p(x)dx \qquad (i)$$

and substituting $\int_{x < t} p(x)dx = 1 - \int_{x \geq t} p(x)dx$ and $\int_{x < t} q(x)dx = 1 - \int_{x \geq t} q(x)dx$ the second inequality becomes

$$
\int_{x \geq t} q(x)dx \int c(x)p(x)dx + \int_{x \geq t} p(x)dx + \int c(x)q(x)dx \geq \\
\int_{x \geq t} p(x)dx \int c(x)q(x)dx + \int_{x \geq t} q(x)dx + \int c(x)p(x)dx.
\qquad (ii)
$$

From the MLRP, $\dfrac{\int_{x \geq t} p(x)dx}{\int_{x \geq t} q(x)dx}$ is increasing in $t$ (by the mediant inequality), so if (i) holds for a given $t_0$, it holds for $t \in [t_0, 1]$. Similarly if (ii) holds for $t_0$, it holds for $t \in [0, t_0]$. Let $t_0$ be the minimum $t$ such that $\int_{x \geq t} p(x)dx = \int c(x)p(x)dx$. At $t_0$, both (i) and (ii) reduce to $\int c(x)q(x)dx \geq \int_{x \geq t_0} q(x)dx$. Writing

$$\int_{x \geq t_0} q(x)dx = \int_{x \geq t_0} (1 - c(x))q(x)dx + \int_{x \geq t_0} c(x)q(x)dx,$$

(i) and (ii) further reduce to $\int_{x < t_0} c(x)q(x)dx \geq \int_{x \geq t_0} (1 - c(x))q(x)dx$. We have

$$\int_{x \geq t_0} p(x)dx = \int_{x \geq t_0} c(x)p(x)dx + \int_{x \geq t_0} (1 - c(x))p(x)dx = \int c(x)p(x)dx$$

so

$$\int_{x < t_0} c(x)p(x)dx = \int_{x \geq t_0} (1 - c(x))p(x)dx.$$

Now, if $\int_{x < t_0} c(x)q(x)dx < \int_{x \geq t_0} (1 - c(x))q(x)dx$, there must be $t_1 \geq t_0$ where the set

$$\{t \mid (t < t_1 \ \& \ p(t_1)q(t) \geq p(t)q(t_1)) \text{ or } (t > t_1 \ \& \ p(t)q(t_1) \geq p(t_1)q(t))\}$$

has full measure and $\int_{x < t_1} c(x)q(x)dx < \int_{x \geq t_1} (1 - c(x))q(x)dx$. Note that if $p(t_1) = 0$ for all such $t_1$, then either $O(P) = 0$, contradicting that $p \succ_{lr} q$, or $O(P) = 1$, contradicting that the ratio $(1 - O(Q))/(1 - O(P))$ is defined, so $t_1$ can be chosen such that $p(t_1) > 0$. But using the MLRP at $t_1$ and that $\int_{x < t_1} c(x)p(x)dx \geq \int_{x \geq t_1} (1 - c(x))p(x)dx$ we have:

$$
\int_{x < t_1} c(x)q(x)p(t_1)dx \geq \int_{x < t_1} c(x)p(x)q(t_1)dx \geq \\
\int_{x \geq t_1} (1 - c(x))p(x)q(t_1)dx \geq \int_{x \geq t_1} (1 - c(x))q(x)p(t_1)dx.
$$

Thus (i) and (ii) must hold at $t_0$, and we therefore have (ii) on $[0, t_0]$ and (i) on $[t_0, 1]$, as required. $\qquad \square$

We note that the lower bounds provided in Lemma 3 are reached in the discrete case when $c(x)$ takes only values 0 and 1, so these bounds are tight in general.

**Definition 5.** The *false negative rate* $FNR_{M,t,P}$ of a model $M$ with decision threshold $t$ on a group $P$ is the proportion of the positive outcomes in $P$ that have predicted value less than $t$. The *false positive rate* $FPR_{M,t,P}$ of a model $M$ with decision threshold $t$ on a group $P$ is the proportion of the negative outcomes in $P$ that have predicted value at least $t$.

The following lemma demonstrates that inequality of these rates across groups is an immediate consequence of monotonic calibration and (strict dominance under) the MLRP.

**Lemma 4.** *Let $p(x)$ and $q(x)$ be the probabilty density functions of the predictions of a model $M$ on two groups $P$ and $Q$. Let $t$ be a decision threshold for the predictions of $M$. If $M$ is monotonically calibrated for groups $P$ and $Q$ (by a function $c(x)$) and $p \succ_{lr} q$, then $FNR_{M,t,P} \leq FNR_{M,t,Q}$ (with equality only if both are 0 or 1) and $FPR_{M,t,Q} \leq FPR_{M,t,P}$ (with equality only if both are 0 or 1).*

*Proof.* Using that $M$ is monotonically calibrated on $P$ and $Q$ the first desired inequality becomes:

$$FNR_{M,t,P} = \frac{\int_{x<t} c(x)p(x)dx}{\int c(x)p(x)dx} \leq \frac{\int_{x<t} c(x)q(x)dx}{\int c(x)q(x)dx} = FNR_{M,t,Q}$$

or equivalently

$$\int_{x<t} c(x)p(x)dx \int c(x)q(x)dx \leq \int_{x<t} c(x)q(x)dx \int c(x)p(x)dx$$

Subtracting $\int_{x<t} c(x)p(x)dx \int_{x<t} c(x)q(x)dx$ from both sides gives

$$\int_{x<t} c(x)p(x)dx \int_{x\geq t} c(x)q(x)dx \leq \int_{x<t} c(x)q(x)dx \int_{x\geq t} c(x)p(x)dx$$

$$\int_{x_0<t} c(x_0)p(x_0)dx_0 \int_{x_1\geq t} c(x_1)q(x_1)dx_1 \leq \int_{x_0<t} c(x_0)q(x_0)dx_0 \int_{x_1\geq t} c(x_1)p(x_1)dx_1$$

$$\int_{x_0<t} \int_{x_1\geq t} c(x_0)c(x_1)p(x_0)q(x_1)dx_1dx_0 \leq \int_{x_0<t} \int_{x_1\geq t} c(x_0)c(x_1)p(x_1)q(x_0)dx_1dx_0$$

which follows directly from the MLRP. To obtain strictness of the inequality, observe that if $p(x_0)q(x_1) = p(x_1)q(x_0)$ almost everywhere on the rectangle $[x_0 < t] \times [x_1 \geq t]$ (which is nontrivial if both FNR values are not 0 or 1), then integrating both sides with respect to $x_0$ over $x_0 < t$ and using the stochastic dominance of $p$ over $q$ gives $p(x_1) = q(x_1)$ almost everywhere on $x_1 \geq t$, and integrating both sides with respect to $x_1$ over $x_1 \geq t$ gives $p(x_0) = q(x_0)$ almost everywhere on $x_0 < t$, contradicting that $p \succ_{lr} q$.

The second desired inequality follows in an identical manner, by reducing

$$FPR_{M,t,Q} = \frac{\int_{x\geq t}(1-c(x))q(x)dx}{\int(1-c(x))q(x)dx} \leq \frac{\int_{x\geq t}(1-c(x))p(x)dx}{\int(1-c(x))p(x)dx} = FPR_{M,t,P}$$

to

$$\int_{x_0<t} \int_{x_1\geq t}(1-c(x_0))(1-c(x_1))p(x_0)q(x_1)dx_1dx_0 \leq$$
$$\int_{x_0<t} \int_{x_1\geq t}(1-c(x_0))(1-c(x_1))p(x_1)q(x_0)dx_1dx_0.$$

$\square$

As with $PPR$ and $PNR$ disparities, $FNR$ and $FPR$ disparities are necessarily small for sufficiently small and large decision thresholds, respectively. The next lemma shows that they are also subject to the same pairwise lower bounds, under the ongoing assumptions.

**Lemma 5.** *Let $p(x)$ and $q(x)$ be the probabilty density functions of the predictions of a model $M$ on two groups $P$ and $Q$. If the model $M$ is monotonically calibrated for groups $P$ and $Q$ (by a function $c(x)$) and $p \succ_{lr} q$, then for any decision threshold $t$ where the ratios below are defined, we have either*

$$\frac{FNR_{M,t,Q}}{FNR_{M,t,P}} \geq \frac{O(P)}{O(Q)}$$

*or*

$$\frac{FPR_{M,t,P}}{FPR_{M,t,Q}} \geq \frac{1 - O(Q)}{1 - O(P)}.$$

*Proof.* Observe that whenever $\int_{x<t} c(x)q(x)dx \geq \int_{x<t} c(x)p(x)dx$, we have

$$\frac{FNR_{M,t,Q}}{FNR_{M,t,P}} = \frac{\int_{x<t} c(x)q(x)dx \int c(x)p(x)dx}{\int_{x<t} c(x)p(x)dx \int c(x)q(x)dx} \geq \frac{O(P)}{O(Q)}.$$

So if the first inequality in the lemma does not hold, we have $\int_{x<t} c(x)q(x)dx < \int_{x<t} c(x)p(x)dx$, from which it follows that $p > q$ almost everywhere in some neighborhood of $t$ (else $p(x) \leq q(x)$ for all $x < t$ from the MLRP). Similarly, whenever $\int_{x\geq t}(1 - c(x))p(x)dx \geq \int_{x\geq t}(1 - c(x))q(x)dx$,

$$\frac{FPR_{M,t,P}}{FPR_{M,t,Q}} = \frac{\int_{x\geq t}(1 - c(x))p(x)dx \int_x (1 - c(x))q(x)dx}{\int_{x\geq t}(1 - c(x))q(x)dx \int_x (1 - c(x))p(x)dx} \geq \frac{1 - O(Q)}{1 - O(P)}.$$

So if the second inequality in the lemma does not hold, $\int_{x\geq t}(1 - c(x))p(x)dx < \int_{x\geq t}(1 - c(x))q(x)dx$, giving $q > p$ almost everywhere in some neighborhood of $t$. Thus at least one of the inequalities holds. □

Both lower bounds in Lemma 5 can be reached simultaneously in the discrete case if $p(x)$ and $q(x)$ disagree only when $c(x) \in \{0, 1\}$, meaning that these bounds are also tight in general. Intuitively, the FNR inequality will fail to hold as $t$ becomes sufficiently large and FNR values approach 1, and the FPR inequality will fail to hold as $t$ becomes sufficiently small and FPR values approach 1. From the first line in the proof, the ratio of FNRs is $\frac{\int_{x<t} c(x)q(x)dx}{\int_{x<t} c(x)p(x)dx}$ higher than the ratio of the outcome rates, and this factor is just the ratio of false negative prevalences within the two populations. Similarly, the relationship of FPRs to the ratio of non-outcome rates is determined by the ratio of false positive prevalences within the two populations. These relationships follow directly from the definitions of FNR and FPR, and Lemma 5 simply shows that, under the given assumptions, at any (non-trivial) threshold the group with a higher outcome rate has either a lower false negative prevalence or higher false positive prevalence. Note that, following the proof, for a threshold $t$ where $q(t) = p(t)$ (and where $t$ is not in the zero-measure set where the MLRP relationship is not satisfied), both inequalities must hold, and in most cases they will both tend to hold in some neighborhood of such a threshold.

**Definition 6.** The *positive predictive value $PPV_{M,t,P}$* of a model $M$ with decision threshold $t$ on a group $P$ is the proportion of the predicted positives (i.e. with predicted value at least $t$) in $P$ that have positive outcome. The *negative predictive value $NPV_{M,t,P}$* of a model $M$ with decision threshold $t$ on a group $P$ is the proportion of the predicted negatives (i.e. with predicted value less than $t$) in $P$ that have negative outcome.

The next lemma shows that these metrics are also unequal across groups under the MLRP and monotonic calibration assumptions, except when the outcome rate or likelihood ratio is (almost everywhere) constant above or below the decision threshold. With any guarantee of strictness in these functions (e.g. strict MLRP and well-calibration), inequality in these metrics is similarly impossible at non-trivial decision thresholds.

**Lemma 6.** *Let $p(x)$ and $q(x)$ be the probabilty density functions of the predictions of a model $M$ on two groups $P$ and $Q$. Let $t$ be a decision threshold for the predictions of $M$. If $M$ is monotonically calibrated for groups $P$ and $Q$ (by a function $c(x)$) and $p \succ_{lr} q$, then for any $t$ where both sides of the inequality are defined,*

- *$PPV_{M,t,Q} \leq PPV_{M,t,P}$, with equality only if $p(x)/q(x)$ or $c(x)$ is constant almost everywhere on $x \geq t$ and*

- *$NPV_{M,t,P} \leq NPV_{M,t,Q}$, with equality only if $p(x)/q(x)$ or $c(x)$ is constant almost everywhere on $x < t$.*

*Proof.* Using that $M$ is monotonically calibrated on $P$ and $Q$ the first desired inequality becomes:

$$PPV_{M,t,Q} = \frac{\int_{x \geq t} c(x)q(x)dx}{\int_{x \geq t} q(x)dx} \leq \frac{\int_{x \geq t} c(x)p(x)dx}{\int_{x \geq t} p(x)dx} = PPV_{M,t,P}$$

or equivalently

$$\int_{x \geq t} c(x)q(x)dx \int_{x \geq t} p(x)dx \leq \int_{x \geq t} c(x)p(x)dx \int_{x \geq t} q(x)dx$$

$$\int_{x_1 \geq t} c(x_1)q(x_1)dx_1 \int_{x_0 \geq t} p(x_0)dx_0 \leq \int_{x_1 \geq t} c(x_1)p(x_1)dx_1 \int_{x_0 \geq t} q(x_0)dx_0$$

$$\int_{x_1 \geq t} \int_{x_0 \geq t} c(x_1)q(x_1)p(x_0)dx_0 dx_1 \leq \int_{x_1 \geq t} \int_{x_0 \geq t} c(x_1)p(x_1)q(x_0)dx_0 dx_1$$

$$0 \leq \int_{x_1 \geq t} \int_{x_0 \geq t} c(x_1)(p(x_1)q(x_0) - q(x_1)p(x_0))dx_0 dx_1$$

If $p(x)/q(x)$ or $c(x)$ is constant almost everywhere for $x \geq t$ the right hand side is 0; otherwise, by the MLRP we have that $p(x_1)q(x_0) - q(x_1)p(x_0) \geq 0$ for almost every pair $x_1 > x_0$ and $p(x_1)q(x_0) - q(x_1)p(x_0) \leq 0$ for almost every $x_1 < x_0$. Viewing the integral as a nonnegative and nonpositive component across these two regions:

$$0 < \int_{x_1 \geq t} \int_{x_1 > x_0 \geq t} c(x_1)(p(x_1)q(x_0) - q(x_1)p(x_0))dx_0 dx_1 + \int_{x_1 \geq t} \int_{x_0 > x_1} c(x_1)(p(x_1)q(x_0) - q(x_1)p(x_0))dx_0 dx_1$$

and interchanging $x_1$ and $x_0$ in the second integral we obtain

$$0 < \int_{x_1 \geq t} \int_{x_1 > x_0 \geq t} c(x_1)(p(x_1)q(x_0) - q(x_1)p(x_0))dx_0 dx_1 - \int_{x_1 \geq t} \int_{x_1 > x_0 \geq t} c(x_0)(p(x_1)q(x_0) - q(x_1)p(x_0))dx_0 dx_1$$

which follows immediately using that $c(x)$ is increasing and nonconstant. The NPV inequality follows in an identical fashion. $\square$

The following theorem unifies the lemmas proven previously, showing that strict inequality in outcome rates or any of the examined metrics implies all of the inequalities or pairs of inequalities given by the earlier lemmas.

**Theorem 1.** *Let $p(x)$ and $q(x)$ be the probabilty density functions of the predictions of a model $M$ on two groups $P$ and $Q$. Let $t$ be a decision threshold for the predictions of $M$ where the ratios in (7)-(10) all exist. If $M$ is monotonically calibrated for groups $P$ and $Q$ and $p$ and $q$ satisfy the MLRP (with either $p \succeq_{lr} q$ or $q \succeq_{lr} p$), then any of (1)-(4) or strict inequality in (5) or (6) implies all of (1)-(6):*

1. *$O(Q) < O(P)$*

2. *$PPR_{M,t,Q} < PPR_{M,t,P}$ (equivalently $PNR_{M,t,P} < PNR_{M,t,Q}$)*

3. *$FNR_{M,t,P} < FNR_{M,t,Q}$ (equivalently $TPR_{M,t,Q} < TPR_{M,t,P}$)*

4. $FPR_{M,t,Q} < FPR_{M,t,P}$ (equivalently $TNR_{M,t,P} < TNR_{M,t,Q}$)

5. $PPV_{M,t,Q} \leq PPV_{M,t,P}$

6. $NPV_{M,t,P} \leq NPV_{M,t,Q}$.

*Any of (1)-(4) or strict inequality in (5) or (6) further implies at least one of (7) or (8), and at least one of (9) and (10):*

7. $\dfrac{PPR_{M,t,P}}{PPR_{M,t,Q}} \geq \dfrac{O(P)}{O(Q)}$

8. $\dfrac{PNR_{M,t,Q}}{PNR_{M,t,P}} \geq \dfrac{1-O(Q)}{1-O(P)}$

9. $\dfrac{FNR_{M,t,Q}}{FNR_{M,t,P}} \geq \dfrac{O(P)}{O(Q)}$

10. $\dfrac{FPR_{M,t,P}}{FPR_{M,t,Q}} \geq \dfrac{1-O(Q)}{1-O(P)}$.

*Proof.* Inequality in any of (2)-(6) implies that $p$ and $q$ are not equal almost everywhere, and the direction of the inequality then implies that $p \succ_{lr} q$. This then implies (1)-(6) from Lemmas 1, 2, 4 and 6, and further implies at least one of (7) and (8) from Lemma 3 and at least one of (9) and (10) from Lemma 5. $\qquad\square$

## 3  COMPAS Analysis

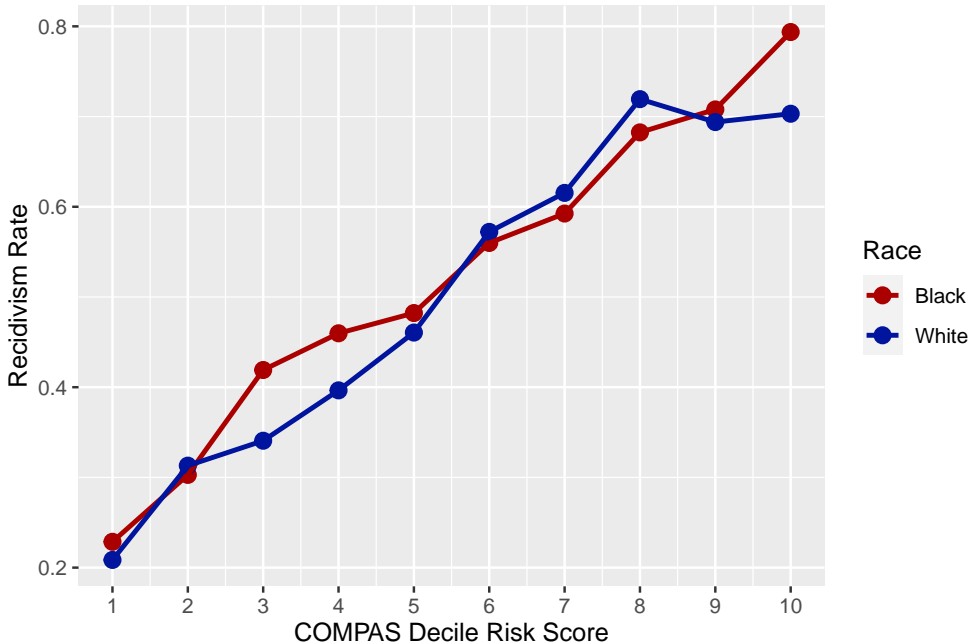

Figure 1: Calibration of the COMPAS risk score

Using ProPublica's COMPAS data (Larson et al., 2016), we observe first that the algorithm is approximately calibrated for the groups of Black and White defendants (Figure 1), and approximately monotonically calibrated for the two groups as well. In addition, viewing the histogram of risk scores across races (Figure 2), we would expect that the distributions of risk scores of these two groups would satisfy the MLRP. Letting $p(x)$

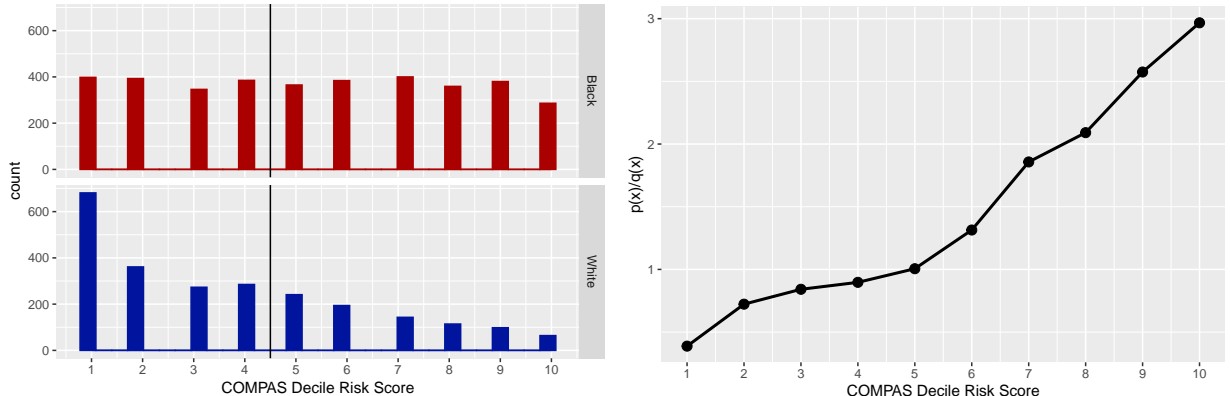

Figure 2: Histogram of the COMPAS risk scores by race

Figure 3: Ratio of likelihood functions for the COMPAS risk scores

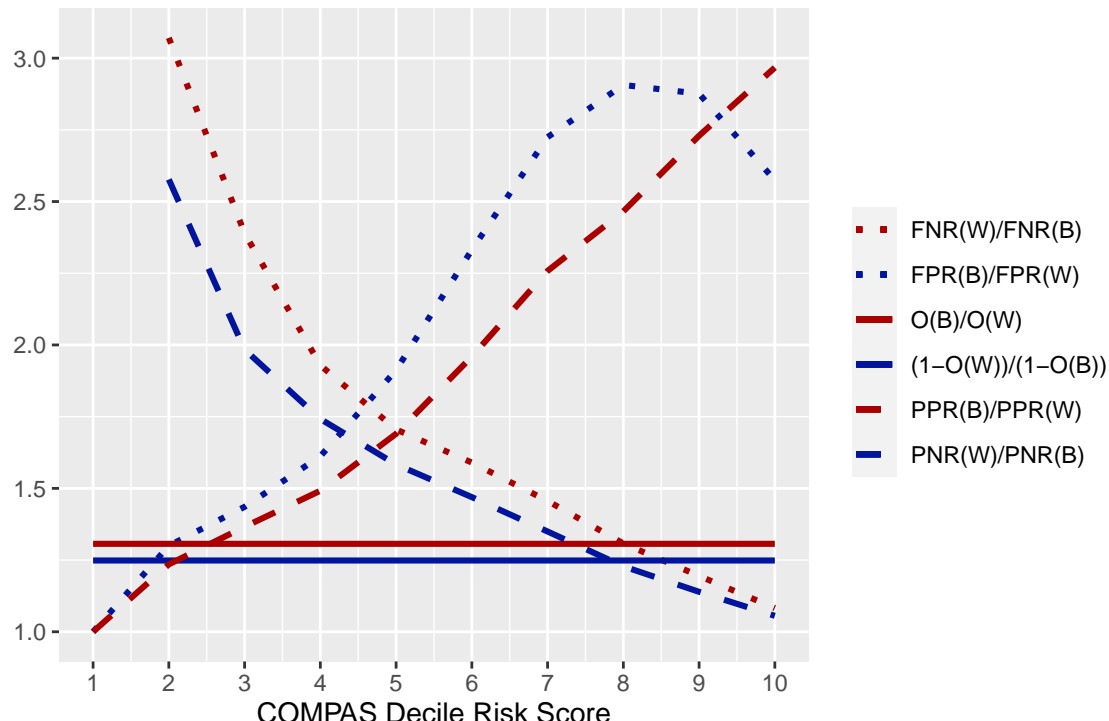

Figure 4: FPR, FNR, PPR, and PNR disparities across different decision thresholds for the COMPAS risk scores

denote the pdf of risk scores for Black defendants and $q(x)$ denote the pdf of risk scores for White defendants, we confirm that $p(x)/q(x)$ is increasing as required (Figure 3). Figure 4 provides the ratios compared in Lemmas 3 and 5 across the possible decision thresholds. The ProPublica analysis used decile scores of 5 or higher (corresponding to "Medium" or "High" risk) as a decision threshold, and because $p(5)$ and $q(5)$ are approximately equal, from the proof of Lemma 5, we can expect both inequalities from Lemma 5 to hold, instead of just one of them. We observe that both inequalities from Lemma 3 hold at this threshold as well.

If significant disparities in false negative/positive rates are guaranteed due to the shapes of the risk distributions of White and Black defendants, is it possible that these differences in distribution, instead of the behavior of the model itself, provide evidence of systemic unfairness? Consider a hypothetical county

with identical distributions of Black and White defendants found in the COMPAS data, except where Black residents are far more likely to be arrested for minor offenses, resulting in the addition of many more lower-risk Black defendants and approximately equalizing the distributions of risk, and therefore the false positive/negative rates, for the resulting Black and White defendants (Corbett-Davies & Goel, 2018). If additionally, White residents are far more likely to not be arrested for minor offenses, we could see the distributions in Figure 2 flip; as a result, in this hypothetical county, the COMPAS algorithm would have a far higher false positive rate for White defendants than for Black defendants. But this would be due to explicit systemic unfairness changing the relative distributions of lowest-risk defendants, where the model's predictions likely have no meaningful impact on the resulting pre-trial decisions, an example of the problem of infra-marginality (Ayres, 2002; Simoiu et al., 2017).

Similar results arise in other contexts, such as the loan approval data examined by Gaebler & Goel (2025). They found that, for a well-calibrated model predicting loan repayment, risk distributions for Black and White borrowers satisfy the MLRP, which would, by Theorem 1, guarantee a higher false negative rate (i.e. more denied loans for applicants who would actually repay) and lower false positive rate (i.e. fewer approved loans for applicants who would not repay) for the group with the lowest repayment rate. A bank that explicitly discriminates against Black borrowers by holding them to a much higher standard for acceptance would see a significantly higher repayment rate for those borrowers, and fail the traditional "outcome test" for discrimination. However, in a setting where data on false negatives is available (e.g. collected from other banks who granted loans to rejected applicants), the model predicting repayment evaluated on that bank's outcomes would have a higher false negative rate for White borrowers, leading one to incorrectly believe the model and/or bank is somehow unfair to White borrowers. Reliance on false positive/negative rates to reveal unfairness in an algorithm or underlying institution can mask, or even further exacerbate, systemic bias.

## 4    Conclusion

We have shown that, in many cases, inequalities of commonly used algorithmic fairness metrics are immediate consequences of differences in outcome rates between groups, and large disparities in at least some of these metrics are guaranteed for groups with large differences in outcome rates. We emphasize that these results should not be the basis for testing group distributions for the MLRP prior to comparing false positive/negative rates or used to help set expected ranges for the disparities in these metrics. Nor should they be viewed as justification for not assessing algorithms for appropriate performance or potential harms across groups, in the numerous ways such harms can arise (Suresh & Guttag, 2021). Instead, they should be taken as further evidence that the specific metrics discussed here, as well as many other metrics commonly used to assess model performance, are highly sensitive to differences in distribution. In contexts where such differences are expected, the ability of these metrics to quantify unfairness seems extremely limited, if not non-existent. While it is straightforward to understand how enforcing demographic parity between groups ignores that different groups may have different outcomes, the same clarity does not appear to exist for false positive/negative rate parity, even though disparities in these metrics are very often due to the same phenomena. Attempts to enforce parity of these metrics will result in sub-optimal decision-making which may harm members of all groups involved (Corbett-Davies et al., 2023), and may actually widen existing disparities. For example, comparing the predictions of a model on two groups under the assumptions of Theorem 1, a group with a significantly lower outcome rate will necessarily have a significantly higher false negative rate for a sufficiently low decision threshold. If the model's intended use is to offer, say, a screening for those at highest risk of a certain medical condition, and the model designer determines that false negative rate parity is a necessary goal and "mitigates" the disparity in the optimal model, the group at highest risk of the condition will be most negatively impacted by the change to the model (i.e most under-screened relative to their actual need). The determination that such a model is always unfair to the lowest-risk group, and increasingly unfair as the risk of this group decreases, is clearly inconsistent with any reasonable notion of fairness.

In the presence of sufficient calibration, disparities in these metrics are direct consequences of differences in group distributions of risk, which are most commonly symptoms of actual differences between groups. Even when these distributional differences are the direct result of systemic inequities, the solution to these

inequities is not to mitigate an error rate difference via intentional mis-calibration or enforce metric parity at the cost of model efficacy (Pfohl et al., 2021). Instead, developers should seek to understand the sources of these differences and ensure their models are ultimately used in ways that close gaps instead of widening them, a process which cannot be reduced to computation and comparison of standard algorithmic fairness metrics.

## 5 Acknowledgments

The author would like to thank Michael Northington V for many helpful discussions during the drafting of this paper and the anonymous reviewers at TMLR for their comments and feedback. The views expressed in this paper are those of the author and not those of his employer.

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
