# OpenReview forum: "Algorithmic fairness with monotone likelihood ratios"
_TMLR — Accepted by TMLR_

### Review · Reviewer_hQ3n · 2025-05-07

**Summary Of Contributions:**

This paper explores the implication of a common fairness criterion for models (monotone calibration, that a model's risk scores correspond to outcomes in the same way across groups) in a decision-making setting with two groups, when the model satisfies the monotone likelihood ratio principle (or MLRP, that risk scores are more likely to be higher for one group than the other). The authors show that these assumptions imply that (a) except in trivial settings, there is a disparity between outcome rates for the two groups and (b) this disparity is a lower bound for the disparity many classic notions of fairness. This implies that in calibrated models that satisfy the MLRP, even approximately guaranteeing other definitions of fairness is fraught because of underlying differences in the outcome rates. The authors then show that in the classic COMPAS dataset, the model's risk scores satisfy the two assumptions (MLRP and monotone calibration).

**Audience:**

Yes

**Claims And Evidence:**

Yes

**Requested Changes:**

In light of the weaknesses identified above, it would be helpful if the authors could
1. further justify the strong MLRP assumption on the model's risk distribution through related work or first principles arguments
2. consider including more text between theorems to improve readability

Possibly I missed this, but from reading the introduction it is unclear that well-calibration is a special case of monotone calibration (although it's clear from the formal definitions). I think this contributes to the strength of your findings, so it could be worth emphasizing this!

**Strengths And Weaknesses:**

### Strengths
- The results are elegant: a fairly simple pair of assumptions on the model yields a variety of novel inequalities.
- Existing work has mainly focused on the impossibility of simultaneously satisfying different definitions of fairness; by establishing _bounds_ this work shows the impossibility of simultaneously _approximately_ satisfying different definitions of fairness.
- The proofs appear to be correct, non-trivial, and well-presented.
- The COMPAS case study helps contextualize the theoretical findings, and shows that the assumptions can hold in natural settings.


### Weaknesses
1. Strength of MLRP assumptions:
	- The paper only references one paper that uses MLRP in the fairness context. While others certainly do so (for example, Chemla and Hennessey [1] and recent work by Chiang et al. [2], it would be helpful to cite other work, as the MLRP assumption is quite strong.
	- Moreover, my sense is that MLRP is more often assumed as a property of the distribution of outcomes rather than the distribution of the model (the paper by Gaebler and Goel cited by the authors does impose this assumption on the risk distribution but my understanding is they assume the model's risk distribution *is* the conditional outcome distribution). If my understanding is incorrect, please correction me! Otherwise, it would be helpful to either identify work that imposes the MLRP assumption on the model's risk distribution, or else further justify why the MLRP should hold for the model's risk scores from first principles.
	- This is especially important for the paper's conclusions, which uses the phrases "in many cases" and "rarely". Without a more solid understanding of the reasonableness of MLRP for the model's risk scores, it is difficult to evaluate these claims.
2. It would be helpful to include more text between theorems to interpret the results in plain language.


### Minor
3. The paper includes formal definitions of calibration and predicted positive rate, but is missing formal definitions of PPV and FPR.


### References
[1] Gilles Chemla and Christopher A. Hennessy. "Controls, belief updating, and bias in medical RCTs." Journal of Economic Theory, 184:104929, 2019. https://www.sciencedirect.com/science/article/pii/S0022053119300808.

[2]  Erica Chiang, Divya Shanmugam, Ashley N. Beecy, Gabriel Sayer, Deborah Estrin, Nikhil Garg, Emma Pierson. "Learning
Disease Progression Models That Capture Health Disparities". AHLI Conference on Health, Inference, and Learning (CHIL), 2025.  https://www.arxiv.org/pdf/2412.16406

---

> ### Author Response · Authors · 2025-05-21
> **Author response**
>
> We thank the reviewer for taking the time to assess the paper and providing constructive comments. We agree with all points raised by the reviewer, and have posted a revised version that addresses the reviewer's concerns as follows:
>
> 1) We have expanded the introduction to provide more examples of the MLRP assumption being used in other contexts, including those referenced by the reviewer (thank you for drawing our attention to these!).
> 2) In the same section we provide more explicit justification for applying the assumption to the model's predictions instead of the underlying risk distributions.
> 3) We reworded the conclusion to clarify our intention behind use of the word "rarely".
> 4) We have added text between lemmas to better explain the results and improve readability.
> 5) We have included formal definitions of FPR, FNR, PPV, and NPV.
>
> If you have any other comments or concerns, please let us know. Thank you again for helping us improve the paper.

---

> > ### Comment · Reviewer_hQ3n · 2025-05-26
> >
> > Thank you for your response!
> >
> > Just to follow up on one point, to address my concern about imposing the MLRP on the model's distribution instead of the underlying risk distribution, you state that "a sufficiently appropriate model will approximate the true risk distributions". I'm confused about this-- aren't these fairness metrics often computed in cases where there is concern that the model risk distribution is *different* from the true risk distribution (e.g., it encodes biases from the training data)?

---

> > > ### Author Response · Authors · 2025-05-27
> > > **Author response**
> > >
> > > The intention of that particular statement is just to provide more justification for why one might expect to encounter risk scores from a model that satisfy the MLRP, i.e. if one could reasonably assume that the underlying group distributions in some setting satisfy the MLRP, then one should expect to encounter (at least some) models in this setting whose risk distributions satisfy the MLRP. We're not suggesting that every model in that setting must satisfy the property, and these results would provide no guarantees about these metrics for those models that do not satisfy the property.
> > >
> > > If we've misunderstood your question, or if you think this could be better clarified in the paper, please let us know. Thank you again for your comments.

---

### Review · Reviewer_mCHv · 2025-05-15

**Summary Of Contributions:**

This paper investigates the compatibility of common algorithmic fairness metrics—such as false positive rate (FPR), false negative rate (FNR), positive/negative predictive value, and selection rate—under the assumptions that: the model is monotonically calibrated, and the group-wise score distributions satisfy the Monotone Likelihood Ratio Property (MLRP).The authors mathematically prove that fairness metric parity is generally impossible across groups with different base outcome rates, even when models are calibrated. They apply this to the COMPAS dataset to show that apparent bias in false positive/negative rates can emerge purely from group-level distribution differences, not from algorithmic unfairness.

**Audience:**

Yes

**Broader Impact Concerns:**

The paper currently lacks a Broader Impact Statement—though touched upon in the conclusion—are not sufficiently addressed. For example, the work can be misunderstood as a justification for ignoring fairness disparities; the paper critiques popular fairness metrics but offers no alternative criteria.

**Claims And Evidence:**

Yes

**Requested Changes:**

A discussion of the strong assumption of MLRP would strengthen the paper. Addressing its practical relevance could enhance the clarity of the contribution. Nonetheless, I find the core findings interesting, and I am inclined to recommend acceptance.

**Strengths And Weaknesses:**

Strengths:

- The introduction of monotonic calibration is both intuitive and practically relevant. The findings on the compatibility of common algorithmic fairness metrics under the assumptions are interesting.
- The mathematical proofs are rigorous and detailed.
- The authors provide a unifying theorem (Theorem 1) that links disparate fairness metrics under MLRP, capturing a broad class of findings.

Weaknesses:

- MLRP is a strong assumption; many real-world models (especially deep learning models) may not satisfy it. Some discussion is provided, but empirical sensitivity analysis is lacking. While the COMPAS case helps, there’s no empirical test across multiple domains to validate how commonly these theoretical conditions apply.

---

> ### Author Response · Authors · 2025-05-21
> **Author response**
>
> We thank the reviewer for their time and thoughtful comments. We agree with all points raised by the reviewer, and have posted a revision that addresses their concerns as follows:
>
> 1) We have expanded the discussion in the introduction to include more background on distributions satisfying the MLRP and references to contexts where the MLRP assumption has been used and/or verified, including empirical analysis by Gaebler & Goel (2025) that found the assumption holding in a variety of domains.
> 2) We have clarified in the conclusion that this should not be viewed as a justification for ignoring fairness disparities (or not even looking for them), only as evidence that certain metrics may not actually detect fairness disparities.
>
> If you have any further concerns or comments, please let us know. Thank you again for helping us improve the paper.

---

### Review · Reviewer_YLGZ · 2025-05-19

**Summary Of Contributions:**

This paper introduces a new theoretical framework for understanding disparities in common algorithmic fairness metrics under realistic modeling assumptions. Specifically, the authors define monotonic calibration, which is a strengthening of traditional calibration that ensures predictions increase with outcome likelihood. They analyze its implications when combined with the monotone likelihood ratio property (MLRP) on group-wise risk distributions. Under these assumptions, they prove that disparities in widely used fairness metrics, such as false positive and negative rates, are not only inevitable when outcome rates differ, but also satisfy tight lower bounds. The authors also provide equivalence relationships among these metric disparities and illustrate their theoretical findings with an empirical analysis of the COMPAS dataset.

**Audience:**

Yes

**Broader Impact Concerns:**

There are no broader impact concerns.

**Claims And Evidence:**

Yes

**Requested Changes:**

-  Provide examples or discussion on when MLRP is realistic in applied settings.
-  Supplement the COMPAS case study with one or two more datasets.

**Strengths And Weaknesses:**

Strengths:
- Addresses widespread misinterpretations of fairness metrics by practitioners and policymakers.
- Moves beyond existing impossibility results by analyzing metric disparities under distributional assumptions rather than just constraints.
- The COMPAS analysis is a compelling illustration that ties the theory back to real-world consequences.

Weaknesses:
- MLRP is a strong assumption that may not hold in many practical cases; more discussion on limitations would be helpful.
- Theoretical results are well-developed, but only a single empirical case study is presented.

---

> ### Author Response · Authors · 2025-05-21
> **Author response**
>
> We thank the reviewer for their helpful comments. We agree with the points raised by the reviewer, and have posted a revision that addresses their concerns as follows:
>
> 1) We have expanded the discussion in the introduction to include more background on distributions satisfying the MLRP and references to contexts where the MLRP assumption has been used and/or verified, including empirical analysis by Gaebler & Goel (2025) that found the assumption holding in a variety of domains.
> 2) We have supplemented the COMPAS case study with a discussion of how disparities could be similarly masked in the loan approval setting studied by Gaebler & Goel.
>
> If you have any further concerns or comments, please let us know. Thank you again for helping us improve the paper.

---

### Comment · Reviewer_mCHv · 2025-05-14

Summary:

This paper investigates the compatibility of common algorithmic fairness metrics—such as false positive rate (FPR), false negative rate (FNR), positive/negative predictive value, and selection rate—under the assumptions that: the model is monotonically calibrated, and the group-wise score distributions satisfy the Monotone Likelihood Ratio Property (MLRP).The authors mathematically prove that fairness metric parity is generally impossible across groups with different base outcome rates, even when models are calibrated. They apply this to the COMPAS dataset to show that apparent bias in false positive/negative rates can emerge purely from group-level distribution differences, not from algorithmic unfairness.

Strengths:
- The introduction of monotonic calibration is both intuitive and practically relevant. The findings on the compatibility of common algorithmic fairness metrics under the assumptions are interesting.
- The mathematical proofs are rigorous and detailed.
- The authors provide a unifying theorem (Theorem 1) that links disparate fairness metrics under MLRP, capturing a broad class of findings.

Weaknesses:

- MLRP is a strong assumption; many real-world models (especially deep learning models) may not satisfy it. Some discussion is provided, but empirical sensitivity analysis is lacking. While the COMPAS case helps, there’s no empirical test across multiple domains to validate how commonly these theoretical conditions apply.

---

### Decision · Action_Editor_SSwV · 2025-07-14

**Recommendation:** Accept as is

**Audience:**

Yes

**Audience Explanation:**

This work studies algorithmic fairness, which is of great interest to the ML community. The specific focus on impossibility results, though no longer a very active area of current research, will be of interest to the TMLR audience. The authors addressed concerns on the strength of the assumption about MLRP, and demonstrated that it is used in practice, and have therefore justified this generalization of results.

**Claims And Evidence:**

Yes

**Claims Explanation:**

This paper generalizes existing impossibility results around fairness in calibrated classification settings. Specifically, the authors study risk models that satisfy the Monotone Likelihood Ratio Property (MLRP), and show that predictions under such a calibration model will suffer some disparity amongst FPR, TPR, FNR, or TNR. The authors provide lower bounds on the levels of disparity.

The results are mathematically rigorous and well presented. The results is a novel generalization of existing fairness results. In the revision, the authors addressed clarity concerns by adding definitions around some of the key terms. Therefore, it meets this criteria for acceptance at TMLR.